# A Novel Framework for Forest Above-Ground Biomass Inversion Using Multi-Source Remote Sensing and Deep Learning

Junxiang Zhang, Cui Zhou *, Gui Zhang, Zhigao Yang, Ziheng Pang and Yongfeng Luo

College of Forestry, Central South University of Forestry and Technology, Changsha 410004, China; 20211100024@csuft.edu.cn (J.Z.); t19861033@csuft.edu.cn (G.Z.); zgyang@126.com (Z.Y.); pang_ziheng@csuft.edu.cn (Z.P.); yfluo@csuft.edu.cn (Y.L.)
* Correspondence: cuizhou@csuft.edu.cn

**Abstract:** The estimation of forest above-ground biomass (AGB) can be significantly improved by leveraging remote sensing (RS) and deep learning (DL) techniques. In this process, it is crucial to obtain appropriate RS features and develop a suitable model. However, traditional methods such as random forest (RF) feature selection often fail to adequately consider the complex relationships within high-dimensional RS feature spaces. Moreover, challenges related to parameter selection and overfitting inherent in DL models may compromise the accuracy of AGB estimation. Therefore, this study proposes a novel framework based on freely available Sentinel-1 synthetic aperture radar (SAR) and Sentinel-2 optical data. Firstly, we designed new indices through the formula analogous with vegetation index calculation to integrate multidimensional spectral and structural information. Then, leveraging the simplicity of computational principles, a pigeon-inspired optimization algorithm (PIO) was introduced into a bi-directional long short-term memory neural network (PIO-BiLSTM), which achieved the set objective function through repeated iteration and validation to obtain the optimal model parameters. Finally, to verify the framework's effect, we conducted experiments in two different tree species and compared another seven classical optimization algorithms and machine learning models. The results indicated that the new indices significantly improved the inversion accuracy of all models in both categories, and the PIO-BiLSTM model achieved the highest accuracy (Category-1: $R^2 = 0.8055$, MAE = 8.8475 Mg·ha$^{-1}$, RMSE = 12.2876 Mg·ha$^{-1}$, relative RMSE = 18.1715%; Category-2: $R^2 = 0.7956$, MAE = 1.7103 Mg·ha$^{-1}$, RMSE = 2.2887 Mg·ha$^{-1}$, relative RMSE = 9.3000%). Compared with existing methods, the proposed framework greatly reduced the labor costs in parameter selection, and its potential uncertainty also decreased by up to 9.0%. Furthermore, the proposed method has a strong generalization ability and is independent of tree species, indicating its great potential for future forest AGB inversion in wider regions with diverse forest types.

**Keywords:** forest above-ground biomass; sentinel; feature combination; BiLSTM neural network; pigeon-inspired optimization

## 1. Introduction

The forest biomass is intricately interconnected with the global carbon cycle and climate variability, and about 70%–90% of biomass is above-ground biomass (AGB) [1–3]. The conventional field measurements of AGB impose limitations on regional coverage and may pose a potentially destructive harvest for forests [4,5]. In contrast, remote sensing (RS) techniques allow for efficient and cost-effective coverage of vast areas, facilitating access to remote regions, and consequently, much research has utilized RS techniques to assist forest AGB inversion [6].

In AGB inversion, RS data selection, RS feature acquisition, and inversion model development are the three most critical steps. Up to now, optical RS data and Synthetic Aperture Radar (SAR) have been widely used in many research studies. Spectral information in



the red and near-infrared bands of optical data is highly sensitive to AGB, which has led numerous studies to utilize vegetation and biophysical parameters for AGB estimation [7,8]. Additionally, SAR can offer valuable insights into dielectric properties and vertical structure by employing appropriate polarization [9–11]. Recently, multi-source data have become a trend in AGB inversion, and previous studies have integrated optical data with SAR data to achieve precise results [12–14]. Within the array of data types, the Sentinel-1 and Sentinel-2 satellites stand out as optimal choices for large-scale and cost-effective AGB mapping, owing to their extensive global coverage and unrestricted accessibility [7].

The acquisition of suitable RS features is a fundamental step in AGB inversion and serves as a prerequisite for modelling. As input data, the availability of Sentinel-1 and Sentinel-2 images makes it possible to produce a large number of RS features; leveraging these features enables the construction of feature sets saturated with multidimensional RS information [7,14,15]. However, excessive features imply information redundancy, which may result in significant disparities between the construction and prediction of models [16]. Consequently, when integrating diverse data sources, it is crucial to acquire the appropriate feature set for AGB inversion. Previous research efforts have primarily focused on selecting those features that are remarkably associated with AGB from the multidimensional RS feature space [17,18]. Jiang et al. [19] analyzed the correlation and calculated the importance degree of each feature using the random forest (RF) algorithm, which consists of numerous decision trees. According to the order of importance from highest to lowest, the features were added sequentially to the feature group for AGB estimation, and the feature group with the highest accuracy was the optimal feature set. Lu et al. [17] argued that the RF algorithm can be employed when the number of sample plots is smaller than independent variables. However, for large and much higher dimensional space, the results are not reliable due to the complex relations in high-dimensional RS feature space and unstable implementation from each decision tree [17,18,20]. Recently, the feature combination in multi-source RS data has become another option for obtaining suitable feature sets. Li et al. [21] constructed a multiplicative model to combine light detection and ranging (LiDAR) with optical data, and thus improved the estimation accuracy of maize crops. Based on the form of the vegetation index formula, Zhang et al. [22] proposed a new feature index that combined optical and LiDAR, which positively impacted AGB estimation. Although this method enables the integration of spectral and structural information to improve the accuracy of AGB inversion, it is mainly applied in optical and LiDAR data sources [21–24]. In addition, the application of large-scale LiDAR data is costly and logistically prohibitive [7]. These limitations have the potential to restrict the application of feature combination, and it is thus necessary to develop an alternative solution for other data sources, such as the freely available and widely archived Sentinel data, which also contain spectral and structural information.

Developing appropriate inversion models is also a crucial step in estimating AGB. Parametric and non-parametric models are the two most commonly used models in AGB inversion [16,25]. The former often includes a limited number of parameters, such as multivariate linear regression (MLR) and generalized linear models (GLM), which require the addition of restrictive hypothesis functions between features and AGB [25,26]. However, due to the inherent complexity in the relationship between AGB and RS data, parametric models often exhibit limited accuracy [27]. Contrary to the parametric models, non-parametric machine learning models process multidimension complex data by adopting more flexible mappings, such as the classical RF and support vector regression (SVR) models [28]. However, for data beyond the boundary of the training samples, RF models usually fail to make accurate predictions, and the performance of SVR models also depends heavily on the choice of kernel function [28,29]. These limitations may lead to a loss of accuracy in AGB estimation. Recently, as a branch of machine learning, deep learning (DL) techniques have become prevalent [30–32]. As an emerging DL model, the bi-directional long short-term memory (BiLSTM) neural network can effectively capture the contextual information in the input feature sequence, which enables it to better understand the pat-

terns and trends of datasets [33–37]. However, due to the complexity of BiLSTM models, human selection relying on experience often fails to result in accurate parameters, which leads to a huge deviation between the accuracy of the training samples and the validation samples, i.e., the occurrence of overfitting phenomenon [38]. Given the specificity of the sample data, all machine learning models have the tendency to overfit during the training process. To tackle these challenges, numerous studies have integrated swarm intelligence algorithms into models [39–41]. In DL modeling, swarm intelligence algorithms are aimed at the optimal solution of parameters and quantify it as the fitness function of population; these algorithms utilize the population's fitness as an indicator for evaluating the results of the optimization process and ultimately obtain the best model parameters by repeatedly and iteratively simulating natural behaviors in populations [42–44]. Compared to classical algorithms such as genetic algorithms (GA), particle swarm optimization (PSO), and whale optimization algorithms (WOA), the pigeon-inspired optimization (PIO) algorithm, as a simulation of pigeon flocks' homing behavior, has a simple computational principle and only requires minimal parameter tuning, which greatly reduces computational expenditure and time cost [45,46].

Based on the aforementioned analysis, this study proposes a novel approach, namely the combined indices optimized inversion framework based on the PIO algorithm and the BiLSTM neural network (CIOPB). Firstly, a novel composite index called combined optical and SAR indices (COSI) has been developed to compensate for the weaknesses in the existing methods. The index is further divided into five groups, along with the original image features to determine the optimal feature group for AGB inversion. Subsequently, to avoid the phenomenon of inaccuracy and overfitting caused by artificial parameter tuning, we perform PIO iterative optimization for the three parameters in the BiLSTM neural network. Specifically, after setting the initial population and the number of iterations, the half mean square error of the predicted response is used as the fitness function, which eventually achieves a high accuracy for both the training data and the testing data by repeated simulation and validation. Finally, the CIOPB framework is utilized to perform AGB estimation in two distinct tree species scenarios and is subsequently compared against other models. Additionally, it is worth noting that, from the AGB calculation of sample plots to the proposed COSI index and inversion model, systematic deviations are inevitably produced throughout the whole workflow, which leads to the uncertainty of results [47,48]. Therefore, this study analyzes these potential challenges and adds the uncertainty of inversion results. In the CIOPB framework, the proposed new indices utilize the global coverage and free accessibility of Sentinel data, which in principle allows for AGB estimation in wider regions. Furthermore, the transferability in DL model enables a promising application of research across different regional environments [49,50].

The structure of this paper is as follows: Section 2 presents the study materials, while Section 3 describes the methodology for AGB inversion. Finally, Sections 4–6 show experimental results, provide discussion, and present conclusions.

## 2. Materials

### 2.1. Test Site

The test site is located in the southeastern part of Yueyang City, Hunan Province, China (113°51′52″–113°58′24″ E, 28°31′7″–28°38′ N, Figure 1), covering a total area of 4762 hectares (ha). The study area is demarcated by higher elevations, and the topography of this area exhibits a gradual elevation gradient, with higher elevations in the southern region gradually descending to lower elevations in the north. The study area falls within the transitional zone from central subtropical to northern subtropical regions and exhibits a humid continental climate with abundant sunshine and precipitation throughout the year. The test site primarily consists of evergreen broad-leaved forests with rich and diverse vegetation species, and there are widespread and structurally complete *Castanopsis eyrei* communities in the area, accompanied by *Cunninghamia lanceolata* and hard broadleaf

species such as *Cyclobalanopsis glauca*, *Quercus serrata*, *Zelkova serrata*, and other species. Consequently, the study area offers favorable site conditions for research purposes.

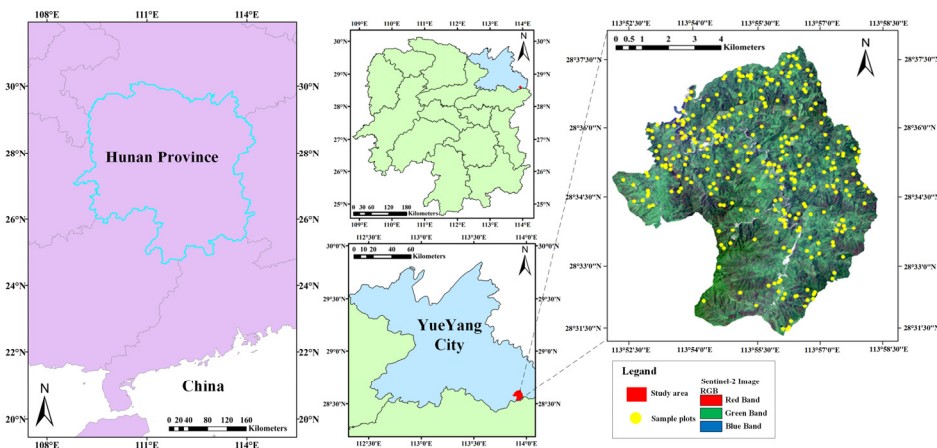

**Figure 1.** Geographical location and RS image of the study area.

### 2.2. Field-Based AGB Calculation

The ground data of this study was obtained from the 2020 Forest Resources Planning and Design Survey of Hunan Province, China, which was sourced from the experimental timberland (Lutou timberland) and stored as vector data. The unit of the ground dataset was small class, which consisted of diverse types of information, including the geographical coordinates of the sample plots, dominant tree species, tree diameter at breast height (DBH), and tree height. According to the sampling design methodology described in the Committee for Earth Observing Satellites (CEOS) AGB validation protocol, a total of 300 forested plots were randomly set up within the timberland boundary, and one plot was taken from each small class [19,27,48]. Based on the available data, the experiment was conducted using two distinct tree species. Simultaneously, trees exceeding a height of 1.3 m and with a DBH greater than 5 cm were carefully selected, and their heights, DBHs, and species were meticulously recorded. By employing the anisotropic growth equations from Table 1 for different groups of tree species [51,52], the AGB for each category was calculated individually to determine the biomass for all sample plots (Table 2).

**Table 1.** The biomass calculation formula for various tree species and groups.

| Type | Tree Species and Group | Biomass Calculation Formula |
|---|---|---|
| Category-1 | Chinese fir wood and other firs | $W_S = 0.0422(D^2H)^{0.8623}$; $W_B = 0.0206(D^2H)^{0.7367}$<br>$W_L = 0.0664(D^2H)^{0.5589}$; $W_T = W_S + W_B + W_L$ |
| Category-2 | Hard broadleaf | $W_S = 0.0545(D^2H)^{0.8630}$; $W_S = 0.0155(D^2H)^{0.8737}$<br>$W_S = 0.0145(D^2H)^{0.7444}$; $W_T = W_S + W_B + W_L$ |

Note: $W_T$, $W_S$, $W_B$, and $W_L$ denote the total AGB value, stem biomass, branch biomass, and leaf biomass, respectively. $D$ is denoted as the DBH (cm), $H$ is denoted as the tree height (m).

**Table 2.** Details for 300 plots.

| Type | Tree Species and Group | Number | AGB (Mg·ha⁻¹) | Mean (Mg·ha⁻¹) | Standard (Mg·ha⁻¹) |
|---|---|---|---|---|---|
| Category-1 | Chinese fir wood and other firs | 170 | 20.57~141.56 | 66.29 | 26.15 |
| Category-2 | Hard broadleaf | 130 | 10.08~40.74 | 24.80 | 4.44 |

*2.3. Remote Sensing Data and Preprocessing*

2.3.1. Optical Data Acquisitions and Preprocessing

The optical image utilized in this study was obtained from Copernicus Open Access Hub (copernicus.eu) of European Space Agency (ESA). The acquisition of the Sentinel-2 L1C-class product occurred on 19 May 2020, which closely aligned with the timeframe of the Forest Resources Planning and Design Survey. Furthermore, the acquired Sentinel-2 images exhibited minimal cloud cover. It is worth noting that Sentinel-2 consists of two satellites, namely 2A and 2B. In this study, we utilized data from the Sentinel-2A satellite, which encompasses 13 operational bands and offers a revisit time of 10 days per individual satellite. Each image covers a swath width of approximately 290 km [53]. Considering that the Sentinel-2 L1C-class product had undergone prior ortho-correction and geometric correction, the subsequent crucial step involved atmospheric correction, which was accomplished using Sen2cor 2.8.0 within the SNAP version 9.0 software. To standardize the resolution of all bands for subsequent analysis, the generated L2A class products were resampled using the bilinear interpolation with a spatial resolution of 10 m [54].

2.3.2. SAR Data Acquisitions and Preprocessing

The SAR images derived from Sentinel-1 data were acquired via the ESA's Copernicus Open Access Hub (copernicus.eu). The specific dates for acquiring the Sentinel-1 images used in this study are 30 August 2020 and 11 September 2020. Equipped with a C-band dual-polarized SAR sensor, the Sentinel-1 satellite is capable of transmitting and receiving signals in both vertical transmit–vertical receive (VV) and vertical transmit–horizontal receive (VH) polarizations [55]. The primary mode of data acquisition utilized in forest AGB studies was the interferometric wide (IW) mode. This method involved employing progressive scan topography observations to cover three sub-regions: IW1, IW2, and IW3. By adopting this approach, consistent image quality was maintained across the entire acquired area [56]. The preprocessing procedure for Sentinel-1 data encompassed handling both single-look complex (SLC) products and multi-look ground range detected (GRD) products. The processing steps for SLC products included orbit correction, radiometric calibration, multi-looking with a Range Looks count of 4 and an Azimuth Looks count of 1, speckle filtering using the Refined Lee filter technique, topography correction, and geocoding. The preprocessing of GRD products also involved thermal noise processing and conversion to decibel scale based on the previous ones. Subsequently, this study employed polarization decomposition to derive the polarization parameters of Sentinel-1 and acquired interferometric parameters using the interferometric synthetic aperture radar (InSAR) technique. These operations were performed utilizing SNAP version 9.0 software.

*2.4. Feature Extraction from Optical and SAR Images*

Table S1 presents all the extracted optical features, including ten vegetation indices and five biophysical parameters. The rationale for selecting these specific vegetation indices is that a wide use of them has been applied in forest AGB estimation, and these indices have a good performance in the process of the fusion of multi-source RS data [22,57]. Additionally, previous studies have demonstrated that these five specific biophysical parameters are effective in reflecting vegetation canopy structure, light energy utilization efficiency, nutrient status of the plants, and overall growth environment, which is also beneficial in AGB inversion [7,58]. In this study, the Biophysical Processor S2 toolbox in SNAP version 9.0 software was utilized to extract five biophysical parameters.

Table S2 presents all the extracted SAR features. Previous studies have demonstrated the potential of backscattering coefficients and polarization decomposition features for AGB inversion [7,57,59]. In this study, we employed the H-Alpha Dual Pol method to conduct polarization decomposition of SLC products, resulting in three polarization parameters: Entropy, Anisotropy, and Alpha angle. Simultaneously, we extracted the incidence angle from the ellipsoid along with two backscatter coefficients, $\sigma_{vh}$ and $\sigma_{vv}$, from GRD products.

Additionally, the interferometric parameters in Sentinel-1 images have also demonstrated a positive impact on AGB inversion [60]. Six major steps were undertaken to acquire the interferometric parameters:

(1) Careful selection and co-registration of images were conducted in this study. The primary image chosen was the Sentinel-1 image acquired on 30 August 2020, while the secondary image selected was obtained on 11 September. Additionally, registration was performed to mitigate errors and rectify pixel-level offsets between the two images.

(2) Phase flattening was conducted in this study. The presence of the flat earth effect can significantly complicate subsequent processing, and thus we required removal of it.

(3) Coherence coefficient calculation and filtering were conducted in this study. A window size of 3 × 3 was utilized for the calculation of coherence coefficients, employing the maximum likelihood method. The Goldstein Phase Filtering technique was applied for the purpose of filtering.

(4) The process of phase unwrapping involves the addition of the wrapped phase to the principal values in order to obtain the true phase.

(5) To mitigate the recurrence of the flat earth effect, refinement and recalibration of orbital parameters were necessary.

(6) The obtained phases measurements were subsequently converted to elevation values, and then geocoded to transform the slant-range coordinate system into a geographic coordinate system. In this study, we obtained phase data, coherence coefficients, and the digital elevation model (DEM).

## 3. Methodology

In order to address the limitation of existing methods, this study proposes a CIOPB framework to estimate AGB. Figure 2 illustrates the process and structure of the CIOPB framework.

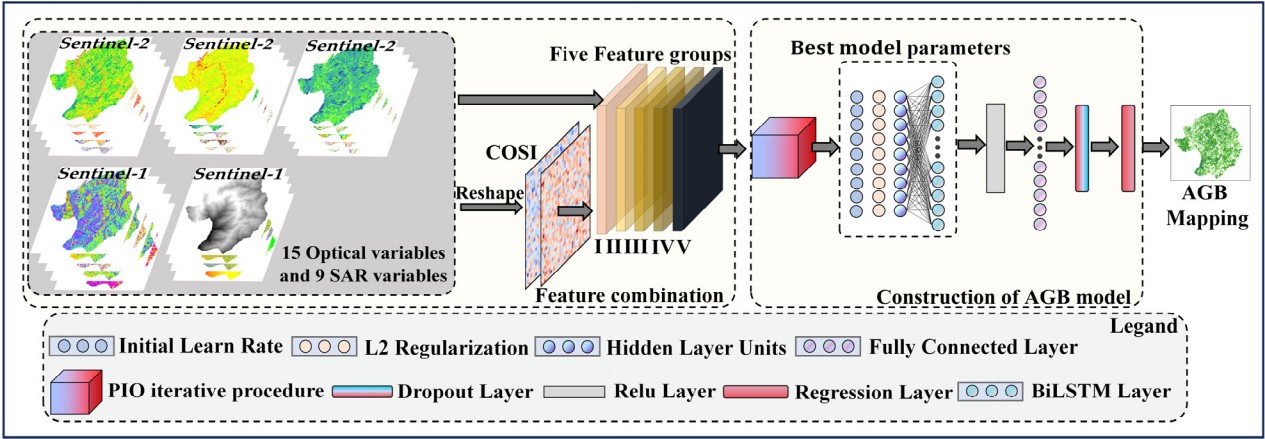

**Figure 2.** Internal structure of the CIOPB framework.

### 3.1. Construction of Feature Set

3.1.1. Definition of COSI Feature Indices

To acquire multi-source features suitable for AGB estimation, Zhang et al. [22] drew inspiration from the principles of vegetation index calculation and devised a novel composite index to effectively integrate LiDAR structural information with Landsat 8 spectral information. Based on the same principle, we present a pioneering index named COSI using freely available Sentinel 1 and Sentinel 2 data, which extends prior investigations on merging optical and LiDAR indices [21–24]. The proposed COSI indices include two distinct types of indices (COSI1 and COSI2), and the formulations are outlined as follows:

$$COSI1 = Optical_i \times SAR_i \tag{1}$$

$$COSI2 = SAR_i\_Optical_i = \frac{SAR_i - Optical_i}{SAR_i + Optical_i} \tag{2}$$

where the SAR variables, represented by $SAR_i$, that exhibit the highest sensitivity to AGB were considered, while $Optical_i$ represents the vegetation indices and biophysical variables. It is important to note that both $SAR_i$ and $Optical_i$ were normalized prior to calculating COSI2 in order to enable a direct comparison of these metrics on a consistent scale and range.

In this study, the sensitivity of RS feature variables to the measured AGB was determined through correlation analysis. Spearman correlation coefficients were calculated for a total of fifteen optical variables and nine SAR variables, which were divided into two categories (Figures S1 and S2). Within Category-1, $\sigma_{vh}$ and $\sigma_{vv}$ emerged as the two SAR variables exhibiting the highest correlation with AGB. Regarding the optical variables, ten vegetation indices were found to meet the requirement perfectly. Additionally, considering their physical significance in the biophysical variables and their relatively high sensitivity to AGB (absolute value of correlation coefficient greater than 0.1), leaf area index (LAI) and fraction of vegetation cover (FVC) were also considered eligible variables. Equations (1) and (2) were utilized to combine $\sigma_{vh}$ and $\sigma_{vv}$ with each of the eligible optical variables.

Similarly, in Category-2, the variables Alpha and Anisotropy in SAR demonstrated the highest correlation with AGB, while only ten vegetation indices exhibited high correlation among the optical variables. Consequently, Alpha and Anisotropy were combined with each of these ten vegetation indices. The results from both categories comprised a total of 88 new indices, as presented in Table S3. The Spearman correlation coefficients of these new indices are displayed in Figure S3.

### 3.1.2. Setting of Feature Combination Scenarios

To obtain an appropriate feature set, the newly acquired COSI indices were integrated with the original RS image variables to generate five distinct feature combination scenarios (Table 3). The rationale for creating these five feature groups is as follows:

(1)   Group I comprised all initial variables extracted from Sentinel data (SV) as an untransformed comparison group to verify that other groups were able to improve the accuracy of AGB inversion.

(2)   Group II encompassed the COSI1 generated using Equation (1) (COSI1), which aimed to validate the performance of the designed COSI1 metric in the AGB inversion.

(3)   Group III included the COSI2 generated using Equation (2) (COSI2), whose purpose was to verify the effectiveness of the designed COSI2 indicator in AGB inversion.

(4)   Group IV consisted of both COSI1 and COSI2 (ACOSI) to explore whether the mutual complementarity of COSI1 and COSI2 can improve the precision of results.

(5)   Group V encompassed all initial variables and the designed indices (AVI), exploring the enhancement effect of the combination of these feature variables.

Subsequently, these five feature groups were sequentially incorporated into the inversion model, whose purpose was to analyze the adaptability of each feature group based on the model's performance. This approach aimed to compensate for the shortcomings of existing methods and provide a reliable outcome for effectively estimating AGB.

**Table 3.** Variables/indices in each feature group.

| Type | Groups | Variables/Indices | Brief Details |
|---|---|---|---|
| Category-1 (Chinese fir wood and other firs) | Group I (SV) | 24 Sentinel variables | All initial Sentinel variables (24) |
| | Group II (COSI1) | 24 COSI1 indices | All COSI1 indices (24) |
| | Group III (COSI2) | 24 COSI2 indices | All COSI2 indices (24) |
| | Group IV (ACOSI) | 24 COSI1 indices and 24 COSI2 indices | All COSI1 and COSI2 indices (48) |
| | Group V (AVI) | 24 Sentinel variables, 24 COSI1 indices, and 24 COSI2 indices | All variables and indices (72) |
| Category-2 (Hard broadleaf) | Group I (SV) | 24 Sentinel variables | All initial Sentinel variables (24) |
| | Group II (COSI1) | 20 COSI1 indices | All COSI1 indices (20) |
| | Group III (COSI2) | 20 COSI2 indices | All COSI2 indices (20) |
| | Group IV (ACOSI) | 20 COSI1 indices and 20 COSI2 indices | All COSI1 and COSI2 indices (40) |
| | Group V (AVI) | 24 Sentinel variables, 20COSI1 indices, and 20 COSI2 indices | All variables and indices (64) |

*3.2. Development of AGB Model*

3.2.1. Bi-Directional Long Short-Term Memory Neural Network

The BiLSTM model and the PIO algorithm are essential components of the inversion model. Previous research has demonstrated the generalizability of the BiLSTM model in prediction tasks, which enables it to extrapolate predictions for new data beyond the training samples [61–63]. As an extended version of long short-term memory (LSTM), BiLSTM consists of both forward and backward LSTM layers. Each LSTM layer is composed of numerous LSTM memory cells, which are utilized to learn the mapping relationship between input RS features and AGBs [33–36]. Figure S4 illustrates the three gates (input, forget, and output gates) in a single LSTM memory cell that collectively determine the state of the LSTM block at runtime. These can be expressed as follows.

$$\begin{cases} i_t = g(W_{ix}x_t + W_{ih}h_{t-1} + b_i) \\ f_t = g\left(W_{fx}x_t + W_{fh}h_{t-1} + b_f\right) \\ o_t = g(W_{ox}x_t + W_{oh}h_{t-1} + b_o) \end{cases} \tag{3}$$

$$c_t = f_t{}^{\circ}c_{t-1} + i_t{}^{\circ}c_t^* \tag{4}$$

$$h_t = o_t{}^{\circ}\tanh(c_t) \tag{5}$$

where $x_t$, $h_t$, $c_t^*$, and $c_t$ represent the input sequence, output sequence, cell activations, and the memory state at time $t$, respectively. The input, forget, and output gates are denoted as $i$, $f$, and $o$, respectively. The symbols $g$, $W$, and $b$ represent the activation function, weight, and bias in each gate, respectively.

The training process of the LSTM is illustrated in Figure S5. The network structure of the BiLSTM hidden layer can be obtained by combining the forward-passing and backward-passing LSTM layers (Figure 3); it can be represented as follows.

$$\overrightarrow{h}_t = LSTM(x_t, \overrightarrow{h}_{t-1}) \tag{6}$$

$$\overleftarrow{h}_t = LSTM(x_t, \overleftarrow{h}_{t+1}) \tag{7}$$

$$y_t = W_{\overrightarrow{h}y}\overrightarrow{h}_t + W_{\overleftarrow{h}y}\overleftarrow{h}_t + b_y \tag{8}$$

where $\overrightarrow{h}_t$, $\overleftarrow{h}_t$, and $y_t$ represent the implicit state of the forward-passing LSTM layers, the implicit state of the backward-passing LSTM layers, and the final output sequence, respectively.

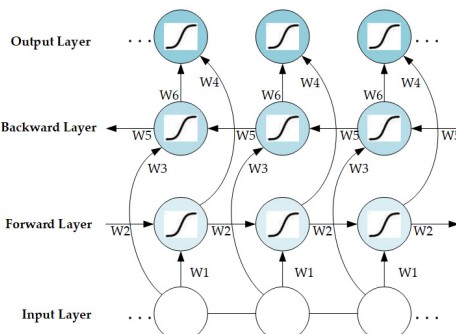

**Figure 3.** Network structure of BiLSTM.

### 3.2.2. Pigeon-Inspired Optimization Iteration Process

The network's structure is primarily determined by the number of LSTM memory cells, making these hidden layer units crucial in the model's performance. However, relying solely on subjective judgment and potential bias from individuals poses challenges in determining the optimal parameters for a model. Moreover, optimal parameters often vary depending on different input feature sets, resulting in significant time costs associated with tuning the model.

This study introduces a PIO iterative process for the automatic parameter computation of BiLSTM (PIO-BiLSTM), aiming to mitigate subjective effects. Compared to other classical optimization algorithms, the PIO algorithm, inspired by the homing behavior of pigeon flocks in nature, has a fast and simple computation with minimal parameter tuning, which greatly reduces the operational cost [45,46]. In Figure 4a, the rightmost grey pigeon represents the optimal flight direction, while thin arrows and thick arrows respectively indicate the previous flight direction and adjusted flight direction.

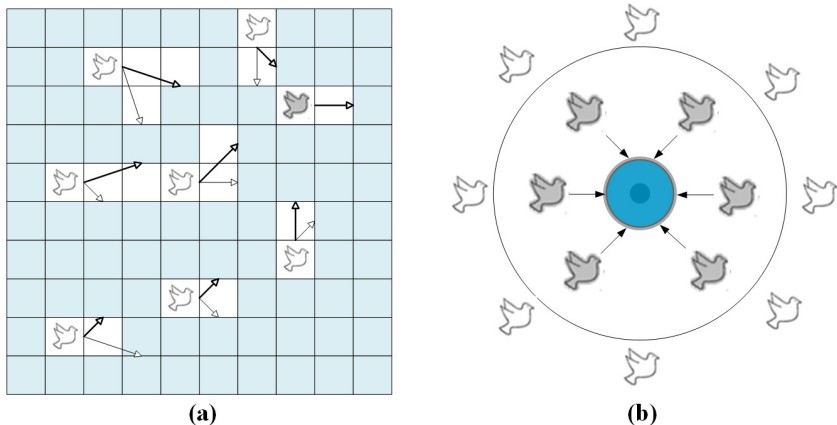

| (a) | (b) |

**Figure 4.** The homing behavior of flocks: (**a**) the PIO's map and compass operator model, (**b**) the landmark operator model.

The position and velocity of the pigeon, denoted as $Pos_i$ and $V_i$, respectively, are update-based on the D-dimensional search space. The state of the $i_{th}$ pigeon at the $t_{th}$ iteration can be expressed as follows.

$$V_i(t) = V_i(t-1) \cdot e^{-Rt} + rand \cdot (Best\_pos - pos_i(t-1)) \tag{9}$$

$$Pos_i(t) = Pos_i(t-1) + V_i(t) \tag{10}$$

where *Best_pos*, *R*, and *rand* represent the current optimal position within the global search space, the influence of map and compass, and a random number, respectively.

The blue circle positioned at the center in Figure 4b signifies the iterative destination of the pigeon flock, indicating that the pigeons within will promptly reach their intended destination.

$$N_p(t) = \frac{N_p(t-1)}{2} \tag{11}$$

$$Pos_c(t) = \frac{\sum Pos_i(t) \cdot fitness(Pos_i(t))}{N_p \sum fitness(Pos_i(t))} \tag{12}$$

$$Pos_i(t) = Pos_i(t-1) + rand \cdot (Pos_c(t) - Pos_i(t-1)) \tag{13}$$

where $N_p$, $Pos_c(t)$, and *fitness* represent the number of flocks reduced per iteration, the center position at the $t_{th}$ iteration, and the quality of the flock, respectively.

The search space in this study was defined as a three-dimensional optimization aimed at determining the optimal values of the three parameters, namely Initial Learn Rate, L2 Regularization, and Hidden Layer Units.

$$Best\_pos = \left[ Best\_pos^{(1)}, \ Best\_pos^{(2)}, Best\_pos^{(3)} \right] \tag{14}$$

The three parameters, meanwhile, were subjected to the physical limitations of the neural network structure and thus required a constraint on their value range.

$$\begin{cases} 1 \times 10^{-10} \ll Best\_pos^{(1)} \ll 1 \times 10^{-2} \\ 1 \times 10^{-4} \ll Best\_pos^{(2)} \ll 2 \times 10^{-3} \\ 10 \ll Best\_pos^{(3)} \ll 100 \end{cases} \tag{15}$$

The loss function of the model was employed as the objective function for PIO iterative optimization, and the $fitness(\cdot)$ in Equation (12) was also computed based on this loss function, which can be expressed as follows.

$$\mathcal{L} = \frac{1}{2} \sum_{i=1}^{N} \left( y_i^{obser} - y_i^{pred} \right)^2 \tag{16}$$

where $y_i^{obser}$, $y_i^{pred}$, and $N$ represent the observed AGB values without back-normalization, the predicted AGB values without back-normalization, and the sample size, respectively.

After iterative updates of the pigeon flock, optimal solutions for these three parameters were obtained in this study. The Hidden Layer Units represented the number of LSTM memory cells, which ultimately determined the network structure of the BiLSTM layer. Subsequently, after passing through multiple network layers of BiLSTM, the input variables were connected to the regression layer to derive predicted values for AGB.

### 3.3. AGB Model Validation

This study employed plots data (N = 74, total in both categories) that were not involved in model construction as independent test samples, and these were randomly selected from the 300 sample plots data in the study area, with the ratio of training data and testing data close to 3:1. The two datasets delineated were not close spatially. To assess the performance of the proposed CIOPB framework, the results of the regression model were quantitatively analyzed using the coefficient of determination ($R^2$), mean absolute error (MAE in Mg·ha$^{-1}$), root mean square error (RMSE in Mg·ha$^{-1}$), and relative RMSE (RMSE$_r$ in %) to assess both accuracy and applicability.

$$R^2 = 1 - \frac{\sum_{i=1}^{n}(y_i - \hat{y}_i)^2}{\sum_{i=1}^{n}(y_i - \overline{y})^2} \tag{17}$$

$$MAE = \frac{\sum_{i=1}^{n} |y_i - \hat{y}_i|}{n} \tag{18}$$

$$RMSE = \sqrt{\frac{\sum_{i=1}^{n} (y_i - \hat{y}_i)^2}{n}} \tag{19}$$

$$RMSE_r = \frac{\sqrt{n^{-1}\sum_{i=1}^{n} (y_i - \hat{y}_i)^2}}{\overline{y}} \tag{20}$$

where $y_i$, $\hat{y}_i$, $\overline{y}$, and n represent the observed AGB value, predicted AGB value, mean value of observed AGB, and number of samples, respectively.

### 3.4. Uncertainty Analysis

The systematic deviations inherent in the AGB inversion framework contribute to the uncertainty in the results. These deviations consist primarily of differences in the measurement of individual wood parameters, biases present in the anisotropic growth model, and errors arising from plot sampling and model predictions [47,64,65]. In this study, the total uncertainty of the AGB estimation consisted of three error sources (these error sources were assumed to be random and independent), which were propagated through the following equations:

$$\varepsilon_{total} = \left( \varepsilon_{measurement}^2 + \varepsilon_{anisotropic\ growth}^2 + \varepsilon_{sampling\ and\ prediction}^2 \right)^{1/2} \tag{21}$$

$$\sigma(s) = CV(s) * \mu AGB \tag{22}$$

$\varepsilon_{measurement}$ represents measurement errors for tree-level parameters averaged at sample plot scale, which were assumed to be 10% based on Chave et al. [66] and Mitchard et al. [67]. $\varepsilon_{anisotropic\ growth}$ represents the bias in calculating the AGB of sample plots using the anisotropic growth model. According to Zhou et al. [51] and Chave et al. [66], this paper set the value of $\varepsilon_{anisotropic\ growth}$ at 11%. The sampling and model prediction error was denoted by $\varepsilon_{sampling\ and\ prediction}$. This study adopted the results from the Guangzhou site of Réjou-Méchain et al. [68], which divided subplots at spatial resolution size in plot data and quantified the local variability using the coefficient of variation (CV) of the subplot AGBs to estimate the sampling error. This site was similar to our study area in terms of climate, topography, and forest types. Subsequently, the standard deviation (SD) of the AGB ($\sigma(s)$) for subplots of spatial resolution size was obtained by multiplying the CV of the subplot ($CV(s)$) by the mean AGB in the plot ($\mu AGB$) through Equation (22). Due to the randomness of the testing set division possibly making the training data spatially close to the testing data [64,69,70], this paper adopted the Monte Carlo method of Li et al. [64] to simulate new residual values, $\varepsilon_\beta$, which were assumed to obey a normal distribution, $\varepsilon \widetilde{N}\left(0, \sigma^2(s)\right)$. Until the results tended to be stable, the $RMSE_r$ after m simulations was used as a metric to quantify the $\varepsilon_{sampling\ and\ prediction}$.

## 4. Results

### 4.1. Performance of the CIOPB Framework in AGB Inversion

Five feature groups were incorporated into the inversion model within the CIOPB framework (Table 4), and the corresponding optimal model parameters are presented in Table S4. Remarkable enhancements in AGB estimation were observed across all categories for the COSI groups (COSI1, COSI2, ACOSI, and AVI) when compared to the initial SV. In Category-1, a significant improvement was noted in ACOSI's $R^2$ value, which increased from 0.6079 to 0.8055 when compared to SV. The MAE decreased by 36.1% (from 13.8437 Mg·ha$^{-1}$ to 8.8475 Mg·ha$^{-1}$), RMSE decreased by 29.4% (from 17.4063 Mg·ha$^{-1}$ to 12.2876 Mg·ha$^{-1}$), $RMSE_r$ decreased by 7.6% (from 25.7413% to 18.1715%), and the uncertainty, $\varepsilon_{total}$, decreased by 6.2% (from 29.7257% to 23.4777%). Additionally, it is worth noting that COSI2 exhibited inferior performance compared to ACOSI, with an $R^2$ value of

0.8042, MAE of 8.7933 Mg·ha$^{-1}$, RMSE of 12.3277 Mg·ha$^{-1}$, RMSE$_r$ of 18.2308%, and $\varepsilon_{total}$ of 23.5236%. The $R^2$ value for AVI in Category-2 exhibited a remarkable improvement from 0.5293 to 0.7956 when compared to SV. Moreover, the MAE decreased by 36.6% (from 2.6972 Mg·ha$^{-1}$ to 1.7103 Mg·ha$^{-1}$), RMSE decreased by 34.1% (from 3.4727 Mg·ha$^{-1}$ to 2.2887 Mg·ha$^{-1}$), RMSE$_r$ decreased by 4.8% (from 14.1109% to 9.3000%), and the $\varepsilon_{total}$ decreased by 3.0% (from 20.4968% to 17.5354%). Notably, ACOSI and AVI were identified as the best feature groups in Category-1 and Category-2, respectively, while COSI1 and COSI2 also demonstrated commendable performance levels. These results validate the significant enhancement of AGB inversion accuracy achieved by the proposed COSI indices, thereby confirming its potential application.

**Table 4.** Performance in five feature groups of the proposed CIOPB framework.

| Method | Type | Feature Group | $R^2$ | MAE (Mg·ha$^{-1}$) | RMSE (Mg·ha$^{-1}$) | RMSE$_r$ (%) | Uncertainty ($\varepsilon_{total}$, %) |
|---|---|---|---|---|---|---|---|
| CIOPB | Category-1 | SV | 0.6097 | 13.8437 | 17.4063 | 25.7413 | 29.7257 |
| | | COSI1 | 0.7275 | 9.8412 | 14.5438 | 21.5081 | 26.1457 |
| | | COSI2 | 0.8042 | **8.7933** | 12.3277 | 18.2308 | 23.5236 |
| | | ACOSI | **0.8055** | 8.8475 | **12.2876** | **18.1715** | **23.4777** |
| | | AVI | 0.7714 | 9.3391 | 13.3197 | 19.6979 | 24.6781 |
| | Category-2 | SV | 0.5293 | 2.6972 | 3.4727 | 14.1109 | 20.4968 |
| | | COSI1 | 0.7426 | 1.7872 | 2.5680 | 10.4348 | 18.1627 |
| | | COSI2 | 0.7081 | 1.9121 | 2.7345 | 11.1113 | 18.5597 |
| | | ACOSI | 0.7091 | 2.0317 | 2.7300 | 11.0930 | 18.5487 |
| | | AVI | **0.7956** | **1.7103** | **2.2887** | **9.3000** | **17.5354** |

Note: The best values of each type of evaluation metric were put in bold.

As illustrated in Table 2, the two categories of tree species in plot data with different AGB distributions and forest age classes. Specifically, Category-1 demonstrates a wider range of AGB distribution (ranging from 20.57 Mg·ha$^{-1}$ to 141.56 Mg·ha$^{-1}$), with a more intricate structure and composition of age classes within forests, whereas Category-2 displays a narrower span of AGB (ranging from 10.57 Mg·ha$^{-1}$ to 40.47 Mg·ha$^{-1}$), indicating a more homogeneous structure and age class composition. These differences between the tree species resulted in divergent performances on the testing data, with Category-1 possessing broader AGB predictions and larger values for the evaluation metrics, while Category-2 showed more aggregated results. For the COSI application, it is worth noting that there was no significant difference in the enhancement between the two tree species, thereby highlighting the applicability of the proposed framework for AGB inversion.

To enhance the visualization of the CIOPB framework's performance in AGB inversion, this study generated scatter plots illustrating the correlation between predicted and observed AGB values (Figure 5). The 1:1 line depicts the congruity between forest AGB predictions and actual measurements. The fitting effect of the four COSI groups was significantly more pronounced compared to SV. Among these groups, ACOSI exhibited the most effective fitting for Category-1, while AVI demonstrated the best fitting for Category-2. The trend lines of these two groups exhibited minimal deviation from the 1:1 line in their respective scatter plots. Moreover, commendable fitting outcomes were also observed in the COSI2 of Category-1 and COSI1 of Category-2. However, it was evident that SV yielded the poorest fitting results in both categories, with predicted values significantly deviating from the observed values at most data points, showing the limitations associated with employing the initial image features from SV as input variables for the inversion model. The above experimental results highlight that the amalgamation of optical and SAR features in COSI resulted in a noticeable reduction in errors between AGB predictions and observations, further substantiating the superior efficacy of COSI in AGB inversion.

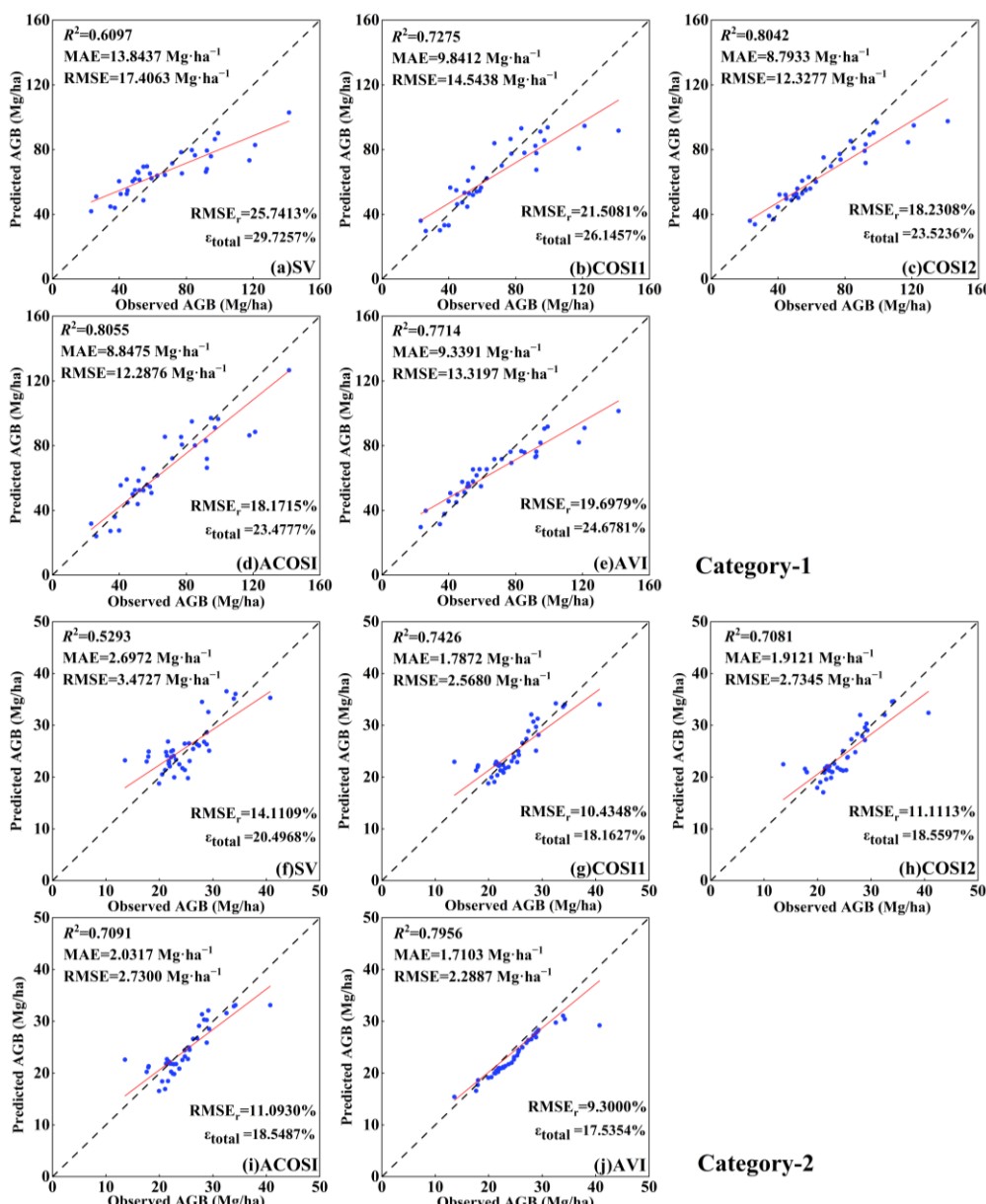

**Figure 5.** Scatterplots of the observed AGB and model-predicted AGB using the proposed CIOPB framework. Note: figures depict the results from five feature groups in two categories, including (**a–e**) Category-1; (**f–j**) Category-2. The blue dots indicate the observed and predicted values for each sample in the testing set; the black line indicates the 1:1 fit line between the observed and predicted values; and the red line indicates the fit trend.

### 4.2. Forest AGB Inversion Mapping

The inversion was conducted for the entire study area based on two categories of experimental tree species, and the spatial distribution reference map of AGB obtained through the CIOPB framework is presented in Figure 6. The total AGB in the study area was estimated to be $5.25 \times 10^5$ Mg, with a mean value of 110.35 Mg·ha$^{-1}$. In order to account for non-forested areas, a few AGB values below 0 Mg·ha$^{-1}$ were considered as underestimations of biomass and were standardized to 0 Mg·ha$^{-1}$ during the mapping process.

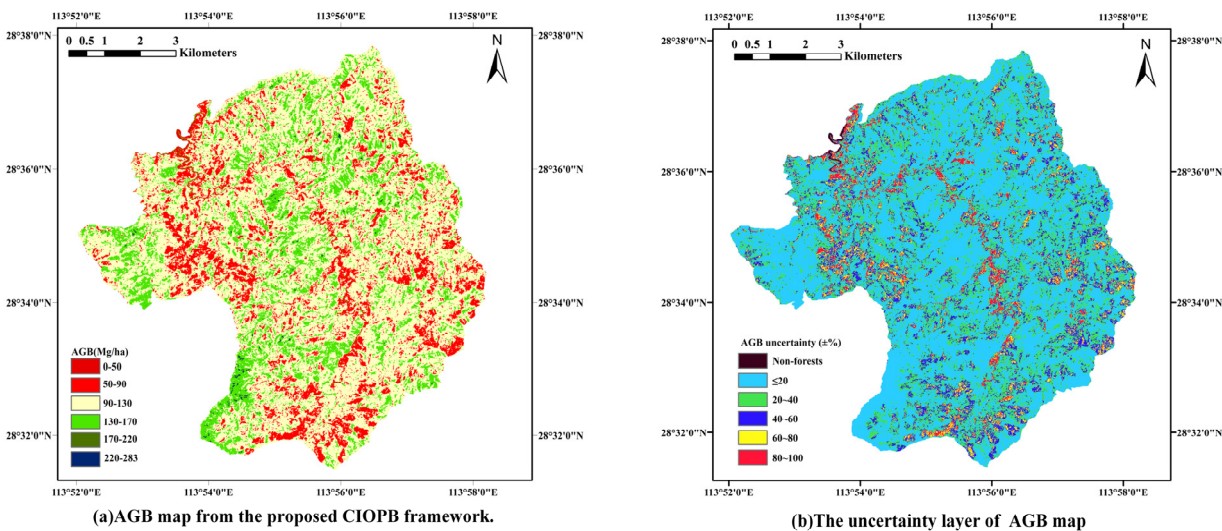

**Figure 6.** Spatial distribution reference map of AGB and the uncertainty layer.

In addition, for the CIOPB framework's predictions across the region, there are still uncertainties in the results due to the series of errors analyzed in Section 3.4. Therefore, this study added an uncertainty layer to the map after integrating the impact of systematic deviations and random errors on the model's predictions. As shown in Figure 6, the uncertainty in the model's predictions for most of the region was located in the 0%–20% and 20%–40% ranges, which roughly matched the results on testing data. For the highest uncertainty ranges, this was mainly due to non-forested areas or other minimal vegetation cover mixed with non-forested cover.

## 5. Discussion

### 5.1. Performance of COSI Indices in Different Models

Within the proposed CIOPB framework, the application of COSI significantly improved the accuracy of the PIO-BiLSTM models. However, it has not been demonstrated whether the application of COSI achieves similar effects in other models. Therefore, to further explore the effect of COSI, all feature groups were also employed to estimate AGB in BiLSTM, LSTM, RF, and SVR models. These four models were selected because the proposed CIOPB framework was constructed on the principles of the BiLSTM and LSTM, while RF and SVR were used as the classical AGB inversion model for comparison. The outcomes were aggregated for comparative purposes (Table S5 and Figure S6).

Results demonstrated that all methods exhibited superior performance in the four COSI groups compared to the initial image features in SV. In both categories, the four COSI groups demonstrated a significant improvement in $R^2$, ranging from 0.106 to 0.266 compared to SV, accompanied by a notable reduction in MAE by 8.8% to 36.6%, a decrease in RMSE by 10.4% to 34.1%, and a decrease in $RMSE_r$ by 2.3% to 7.6%. For all methodologies considered, ACOSI and AVI remained as the optimal feature group for Category-1 and Category-2, respectively. Additionally, COSI was also effective in reducing model uncertainty. These observations demonstrated that the COSI indices significantly improved the accuracy of inversion in each model, regardless of the specific inversion model used. Furthermore, there was no significant difference in performance improvements between the two categories of tree species, thus further validating its potential for AGB inversion. The findings are consistent with previous research on optical and LiDAR data, where the incorporation of combined indices has consistently led to improved accuracy in estimating AGB [21–24]. However, the integration of optical and SAR data has been rarely explored in these studies, presenting an opportunity for our research.

## 5.2. Accuracy Comparison Using Different Models in AGB Prediction

To validate the proposed PIO-BiLSTM model, the optimal feature group was incorporated into the other models to conduct comparative experiments (refer to Table 5 and Figure 7). Specifically, the BiLSTM models based on GA, PSO, and WOA algorithms (GA-BiLSTM, PSO-BiLSTM, WOA-BiLSTM) were used to validate the performance of the selected PIO iterative process; BiLSTM was introduced as the base model to explore whether the optimization algorithms have the effect of improving the model's accuracy, and LSTM, RF, and SVR were used as the classical machine learning models for comparison.

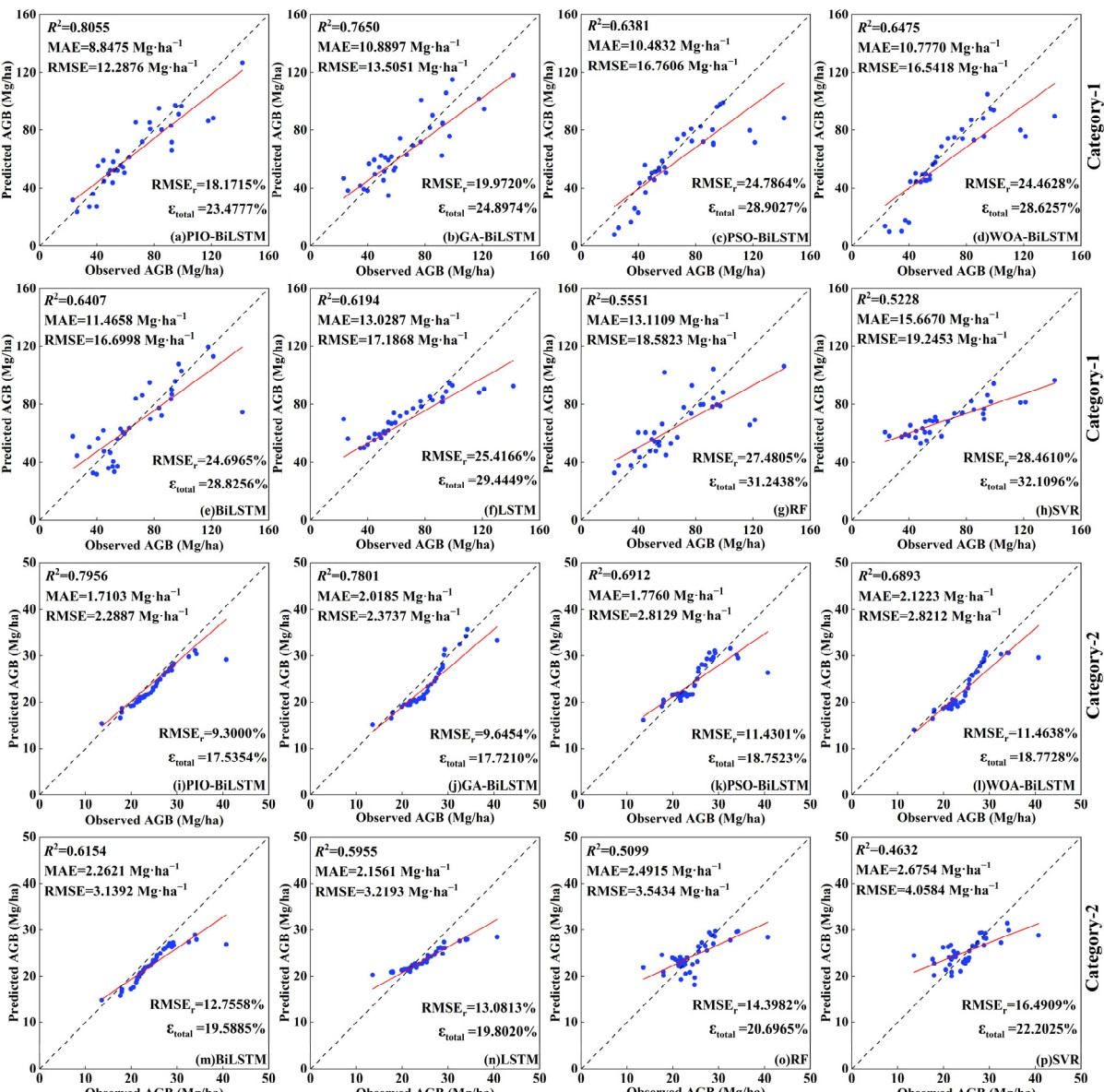

**Figure 7.** Scatterplots of the observed AGB and model-predicted AGB using the eight methods in two categories. Note: figures depict the different inversion models from the best feature group in two categories, including (**a**,**i**) PIO-BiLSTM, (**b**,**j**) GA-BiLSTM, (**c**,**k**) PSO-BiLSTM, (**d**,**l**) WOA-BiLSTM, (**e**,**m**) BiLSTM, (**f**,**n**) LSTM, (**g**,**o**) RF, and (**h**,**p**) SVR. The blue dots indicate the observed and predicted values for each sample in the testing set; the black line indicates the 1:1 fit line between the observed and predicted values; and the red line indicates the fit trend.

**Table 5.** Performance of the eight models from the best feature group in two categories.

| Type | Methods | $R^2$ | MAE (Mg·ha$^{-1}$) | RMSE (Mg·ha$^{-1}$) | RMSE$_r$ (%) | Uncertainty ($\varepsilon_{total}$, %) |
|---|---|---|---|---|---|---|
| | **PIO-BiLSTM** | **0.8055** | **8.8475** | **12.2876** | **18.1715** | **23.4777** |
| | GA-BiLSTM | 0.7650 | 10.8897 | 13.5051 | 19.9720 | 24.8974 |
| | PSO-BiLSTM | 0.6381 | 10.4832 | 16.7606 | 24.7864 | 28.9027 |
| Category-1 | WOA-BiLSTM | 0.6475 | 10.7770 | 16.5418 | 24.4628 | 28.6257 |
| | BiLSTM | 0.6407 | 11.4658 | 16.6998 | 24.6965 | 28.8256 |
| | LSTM | 0.6194 | 13.0287 | 17.1868 | 25.4166 | 29.4449 |
| | RF | 0.5551 | 13.1109 | 18.5823 | 27.4805 | 31.2438 |
| | SVR | 0.5228 | 15.6670 | 19.2453 | 28.4610 | 32.1096 |
| | **PIO-BiLSTM** | **0.7956** | **1.7103** | **2.2887** | **9.3000** | **17.5354** |
| | GA-BiLSTM | 0.7801 | 2.0185 | 2.3737 | 9.6454 | 17.7210 |
| | PSO-BiLSTM | 0.6912 | 1.7760 | 2.8129 | 11.4301 | 18.7523 |
| Category-2 | WOA-BiLSTM | 0.6893 | 2.1223 | 2.8212 | 11.4638 | 18.7728 |
| | BiLSTM | 0.6154 | 2.2621 | 3.1392 | 12.7558 | 19.5885 |
| | LSTM | 0.5955 | 2.1561 | 3.2193 | 13.0813 | 19.8020 |
| | RF | 0.5099 | 2.4915 | 3.5434 | 14.3982 | 20.6956 |
| | SVR | 0.4632 | 2.6754 | 4.0584 | 16.4909 | 22.2025 |

Note: The best values of each type of evaluation metric were put in bold.

Compared to other models, the proposed PIO-BiLSTM model achieved significantly higher accuracy in AGB inversion (Category-1: $R^2$ = 0.8055, MAE = 8.8475 Mg·ha$^{-1}$, RMSE = 12.2876 Mg·ha$^{-1}$, RMSE$_r$ = 18.1715%; Category-2: $R^2$ = 0.7956, MAE = 1.7103 Mg·ha$^{-1}$, RMSE = 2.2887 Mg·ha$^{-1}$, RMSE$_r$ = 9.3000%), indicating a greater level of consistency between predicted and observed AGB. In these results, there was an improvement ranging from 0.0280 to 0.3076 in the average $R^2$ values for two tree species, while the average MAEs for two tree species decreased by 17.0% to 39.8%, the average RMSEs decreased by 6.3% to 39.9%, the average RMSE$_r$s decreased by 1.1% to 8.7%, and the average uncertainty for two tree species decreased by 0.8% to 9.0% compared to other models. For the other three classical optimization algorithms, the GA-BiLSTM model showed the smallest gap compared with PIO-BiLSTM, but the GA optimization process has paid a huge computational expenditure in AGB inversion. Notably, the accuracy of PSO-BiLSTM and WOA-BiLSTM did not deviate much from the BiLSTM and LSTM models, which fitted well to the middle range of AGB values but generated some underestimates at both high and low values. The BiLSTM and LSTM models faced challenges in parameter selection and overfitting during training. Only by utilizing the parameters obtained from the CIOPB framework (Table S4) and fine-tuning them could a relatively good inversion accuracy be obtained. These results demonstrate that the optimization process can significantly reduce the cost and uncertainty in model parameter selection. Moreover, it was obvious that both the RF and SVR models demonstrated the worst model performance, which presented extremely significant deviations in scatterplot analysis. The superiority of DL models over machine learning models in this study can be attributed to their multi-layer neural network structure, which facilitates the systematic learning of features and patterns within data [16,71]. Furthermore, because the base units in BiLSTM and LSTM vary with each time step, there are distinct strengths of both models in dealing with time series [33–36]. Ge et al. [33] demonstrated the feasibility of LSTM models for forest height time-series monitoring, which enables a promising prospect in biomass monitoring changes.

Moreover, both machine learning and DL models have a good potential for tasks in other areas [49]. However, for the transferability of AGB models, there have been fewer studies on it [50]. In future research, we aim to consider employing more DL models with swarm intelligence algorithms and validate the transferability of these models with additional study areas and tree species to compensate for the limitations of our current work.

*5.3. Uncertainty of AGB Inversion Using the CIOPB Framework*

The proposed CIOPB framework has achieved significant enhancements in terms of both features and models. However, the systematic deviations inherent in the AGB inversion workflow enable the framework to still leave room for further improvement. In the CIOPB framework, uncalibrated measurement tools and human error allow tree-level measurement errors to propagate into the anisotropic growth model [47]; furthermore, the randomness in plot sampling and testing set delineation is also an important error source, which may generate potential uncertainty in prediction results. Based on the methods and results of previous studies [47,64–68,72,73], this paper analyzed the uncertainty generated in the CIOPB framework (Section 3.4) and added the uncertainty layer of the output AGB map (Figure 6) in a relatively ideal state [74]. The results revealed that the uncertainty of the CIOPB framework on the testing set samples ranged from 17.5354% to 29.7257% (Category-1: 23.4777% to 29.7257%; Category-2: 17.5354% to 20.4968%), with lower uncertainty using the optimal feature group than other groups. Compared to other models, the PIO-BiLSTM model in this framework showed the lowest uncertainty, and the average uncertainty for two tree species decreased by 0.8% to 9.0%. Furthermore, the AGB spatial distribution reference map produced by the CIOPB framework is also subject to uncertainty due to the above-mentioned errors. Our result showed that the uncertainty in most regions lies in the 0%–20% and 20%–40% intervals, which was approximately in line with the performance on the testing set; the majority of the regions with extremely low AGB have the highest uncertainties, which was consistent with Rodríguez-Veiga et al. [65]. Their research utilized the maximum entropy (MaxEnt) algorithm to estimate AGB, uncertainty, and forest probability, which provided a more accurate spatial distribution compared to existing AGB map products [65]. For the uncertainty of results, most of these originate from the systematic deviation of plot-level data, which are hard to quantify intuitively due to the complexity of their sources. Moreover, in machine learning, there is always only a single testing dataset to evaluate the accuracy of AGB estimation results, while ignoring the residual variability produced by the model. The analysis of these errors is precisely lacking in most of the current AGB estimation studies, leading to potentially unreliable modeling results. In order to minimize the impact of uncertainty on the results, a comprehensive analysis of the errors in the AGB inversion workflow is a necessity for future research to improve the inversion accuracy.

## 6. Conclusions

The significance of forest AGB in the global carbon cycle and its contribution to mitigating climate change cannot be overstated. RS and DL technologies play a pivotal role in achieving precise and efficient inversion of forest AGB. However, there are unresolved issues with the current methodology, such as limitations on the acquisition of suitable feature sets, challenges associated with selecting model parameters, and susceptibility to overfitting. Therefore, a CIOPB framework integrating Sentinel-1 and Sentinel-2 images was proposed to address the existing challenges in the construction of feature sets and models for AGB estimation.

The proposed methodology encompasses the acquisition of feature sets and the construction of an inversion model within a unified framework. Specifically, we established an innovative index that combines optical and SAR data to identify the most effective feature group. Additionally, we incorporated a PIO iterative process into the BiLSTM neural network to determine optimal parameters, effectively addressing challenges related to parameter selection and overfitting. To evaluate the performance of this framework, we conducted comparative analyses using other classical optimization algorithms and machine learning models and discussed the systematic deviations with uncertainty generated in the workflow. All methods were employed for estimating AGBs in two tree species with different compositions and structures. The following conclusions were drawn:

(1) The incorporation of the designed COSI indices significantly enhanced the accuracy of forest AGB inversion by constructing suitable feature sets. When considering the

influence of different tree species and inversion models on features, COSI indices consistently outperformed the initial image-extracted features. The combination of ACOSI demonstrated an appropriate feature group in Category-1, while utilizing AVI proved to be the most effective feature group in Category-2.

(2) The PIO-BiLSTM model demonstrated superior performance in two categories (Category-1: $R^2$ = 0.8055, MAE = 8.8475 Mg·ha$^{-1}$, RMSE = 12.2876 Mg·ha$^{-1}$, RMSE$_r$ = 18.1715%; Category-2: $R^2$ = 0.7956, MAE = 1.7103 Mg·ha$^{-1}$, RMSE = 2.2887 Mg·ha$^{-1}$, RMSE$_r$ = 9.3000%) when compared to the GA-BiLSTM, PSO-BiLSTM, WOA-BiLSTM, BiLSTM, LSTM, RF, and SVR models. The average MAEs decreased by 17.0% to 39.8% and the average RMSEs decreased by 6.3% to 39.9%, while the average RMSE$_r$s decreased by 1.1% to 8.7%. Additionally, the average uncertainty for two tree species decreased by up to 9.0% compared to the other models. It is worth noting that there was no significant difference in the enhancement effect of the PIO-BiLSTM model between these two categories of trees.

Compared to existing methods, the CIOPB framework has demonstrated commendable results in achieving cost-effective and precise AGB inversion for subtropical forests, and it also has great potential for application in different environmental regions and AGB change monitoring. Moving forward, we aim to fully leverage the proposed methodology by validating it on a larger scale encompassing diverse tree species and forest types, while also integrating cutting-edge algorithms to optimize computational efficiency.

**Supplementary Materials:** The following supporting information can be downloaded at: https://www.mdpi.com/article/10.3390/f15030456/s1, Table S1. Feature extraction from the Sentinel-2 image. Table S2 Feature extraction from the Sentinel-1 image. Table S3 Newly generated COSI indices. Table S4 The optimal parameters in the CIOPB framework. Table S5. Performance of five models in all feature groups of two categories. Figure S1. Heat map of absolute values for correlation coefficients between remote sensing variables and AGB in two categories: (a) Category-1; (b) Category-2. Figure S2. Histogram of absolute values for correlation coefficients between remote sensing variables and AGB in two categories: (a) Category-1; (b) Category-2. Figure S3 (a–d) The histogram of the absolute values for the correlation coefficients of the new generated indices in two categories:(a) Combined with $\sigma_{vh}$ in Category-1, (b) Combined with $\sigma_{vv}$ in Category-1, (c) Combined with Alpha in Category-2, (d) Combined with Anisotropy in Category-2. Note: The numbers in the horizontal coordinate represent, in turn, the new indices in Table S1. Figure S4. Construct of a single LSTM memory cell. Figure S5 Training process of the LSTM. Figure S6. Comparison of model evaluation metrics using the SVR, RF, LSTM, BiLSTM, and the PIO-BiLSTM in two categories: (a) Category-1; (b) Category-2.

**Author Contributions:** Conceptualization, J.Z. and C.Z.; methodology, J.Z., C.Z. and G.Z.; validation, J.Z., C.Z., G.Z., Z.Y., Z.P. and Y.L.; formal analysis, J.Z., C.Z. and G.Z.; investigation, J.Z., C.Z., G.Z., Z.Y. and Y.L.; resources, Z.P.; data curation, G.Z.; writing—original draft preparation, J.Z.; writing—review and editing, J.Z. and C.Z.; supervision, C.Z.; project administration, C.Z., G.Z., Z.Y. and Y.L.; funding acquisition, C.Z. All authors have read and agreed to the published version of the manuscript.

**Funding:** This research was funded by the National Natural Science Foundation of China (No. 42074016 and 42030112).

**Data Availability Statement:** The Sentinel-1 and Sentinel-2 data that support the findings of this study are available at (Open Access Hub (copernicus.eu)). The forest resources survey database data that support the findings of this study are available on request from Junxiang Zhang (20211100024@csuft.edu.cn). The data are not publicly available due to privacy.

**Acknowledgments:** Many thanks to NASA, USGS for generously providing the datasets free of charge.

**Conflicts of Interest:** The authors declare that they have no known competing financial interests or personal relationships that could have potentially influenced the work reported in this paper. The authors declare no conflicts of interest.

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
