# Peer review of "A Novel Framework for Forest Above-Ground Biomass Inversion Using Multi-Source Remote Sensing and Deep Learning"

_forests, doi:10.3390/f15030456_

Round 1

Reviewer 1 Report

Comments and Suggestions for Authors

Rephrase line 10 .. should be "plays a pivotal role in climate change mitigation." Line 11 appear out of nowhere. Briefly but clearly define feature set .

Line 14 "constrained by specific application scenarios." is vague

Novel and robust framework is hard to justify. 

Line 35 "with approximately 70–90% of it comprising aboveground biomass" rephrase this. 

"pose environmental disruptions to" You mean destructive harvest? Clarify!

Line 49 Can you put more context first for "inversion" and "construction of features"

Rephrase line 50-51 if you're simply saying SEntinel-1 and 2 are input data to produce "feature variables"

I'm really confused how you contextualized feature set. Say if you put a mix of nonsense features (noise) and sensitive features (Sentinel bands), the model would "react" by itself e.g. by showing variable importance. The greater concern is how correlated the feature set is. I'm sorry I cannot understand lines 57 onwards the end of the paragraph

line 69 needs a reference

rephrase "hypothesis-driven processes"

I can't understand this "requires assumptions that are based on the underlying data distribution."

machine learning is non-parametric right? Not just LSTM 

"bi-directional information from both past and future time steps. This enables it to overcome common challenges such as gradient vanishing and explosion that often arise when applied to prediction tasks" rephrase. You should write as if your neighbor is reading your paper and assume your neighbor is stupid

Line 83.. All machine learning models are prone to overfitting 

Can you provide more literature with regards to the applicatoin of BiLSTM to mapping of any environmental variable especially AGB?

Give more context to PIO

2.2 can you talk about the sampling design of the field plots?

Did you consider using a uniform data acquisition workflow e.g. using GEE?

So you did InSAR meaning you have phase and coherence data?

Figure 2 I can't read the texts inside the box

Add a columns of Table 3 providing brief details about the category and groups, Can you also name the groups by the models used themselves?

Why the authors did not compare BiLSTM to the LSTM itself? 

Can BiLSTM extrapolate predictions meaning it can predict reasonably well outside the range of the training data?

3.3 I strongly suggest to include Mean error and an efficiency metric like NSE, see de Bruin et al. 2022 Ecological Indicators

Clearly elaborate what dataset did you use for validation? Are they close with the training data spatially? 

I wont pay attention much to R2 and RMSE as these are indications of dispersion, random error. Systematic deviaitons, on the other hand, should be analyzed and discussed see Araza et al. 2022 A comprehensive framework for assessing the accuracy and uncertainty of global above-ground biomass maps - ScienceDirect

Revise Figure 7 to a more uniform AGB binning e.g. 0-10, 10-25, 25-50, ... reverse the colors also of course the greener the more biomass! Can you manage to produce an uncertainty layer of this map? 

I would rename CIOPB into a more optimized-sounding deep learning model.. e.g. PioLSTM something like that

So in Category 2, it appears data are aggregated 0-50 Mg/ha range. Can oyu provide a map showing predictions of Category 1(best model) and Category 2(best model). It wasn't clear to me whether the optimization also leads to aggregatoin?

In the discussion, can you say somehting about the transferability of the model e.g. into other (similar) regions? upscaling potential.. processing power and time demand

Also working with time series remote sensing data. I htink LSTM models are good at working at such. Can you also add its potential for biomass change monitoring?  

Comments on the Quality of English Language

Introduction needs imporvement. There's a lot of technical terms that need to be contextualized first. Sentences connection also needs improvemnet. 

Author Response

Dear Reviewer 1

Thank you very much for the time and effort to have our manuscript reviewed, and for giving us the opportunity to revise it. The referees’ comments/suggestions are very helpful to further improve this manuscript. We have fully revised it according to them. For your convenience, the main revisions are marked with blue font in the forests-2850118-ori 2.17- revised. The point-to-point responses to the comments are in the author-coverletter-34775830.v1.docx

Best regards

Yours sincerely,

Cui Zhou, Ph.D. & Professor

Central South University of Forestry and Technology, Changsha Hunan, 410004, China,

Reviewer 2 Report

Comments and Suggestions for Authors

Dear Authors

The research titled “A novel framework integrating multi-source remote sensing  feature and deep learning model for forest aboveground biomass inversion” proposes reated new feature indices, combining valuable data from optical and synthetic aperture radar sources. Additionally, this study incorporated a Pigeon-inspired iteration process into the Bi-directional Long Short-Term Memory (BiLSTM) neural network to fine-tune the model's parameters. The manuscript seems to have some valuable insights; however, certain aspects might be considered as potential issues:

1.     The title is quite long. This complexity might make it less accessible to a general audience and could be challenging for search engine optimization.

2.     Extensive English editing is required.

3.     There are a few issues in the abstract as follows:

The abstract lacks a clear and specific statement of the problem or gap in existing methods that the proposed framework aims to address.

While mentioning challenges in DL models and conventional methods, the abstract does not provide detailed insights into these challenges, limiting the reader's understanding.

The abstract briefly mentions the design of new multi-source feature indices but lacks detailed information on their nature, creation process, or the rationale behind their selection.

The introduction of a Pigeon-inspired iteration process is mentioned, but its significance, advantages, or how it specifically contributes to optimizing the BiLSTM model's parameters is not clarified.

While the abstract highlights the improved accuracy of the proposed framework, it lacks a comparative analysis with other relevant state-of-the-art models in the field.

There is no discussion on the generalizability of the proposed method to different forest types, geographic regions, or environmental conditions.

The abstract does not explicitly acknowledge any limitations or potential drawbacks of the proposed framework, which is essential for a comprehensive understanding of the study.

The abstract could provide more context on how the proposed method contributes to the broader field of forest aboveground biomass inversion and its potential applications beyond the specific experimental scenarios.

The abstract presents results in terms of reduction percentages without contextualizing them with the actual values, making it challenging to interpret the practical significance of these reductions.

4.     Introduction

The latest works should be added as [1-3].

The introduction primarily focuses on specific scenarios, potentially limiting the broader applicability of the proposed framework.

Although the importance of an appropriate feature set is acknowledged, the introduction lacks a detailed exploration of challenges or potential drawbacks related to the chosen "combined optical and SAR indices (COSI)" method.

The introduction introduces the CIOPB framework but does not delve into alternative models, limiting a comprehensive understanding of the best approach.

The introduction briefly mentions addressing overfitting challenges but lacks in-depth details on how the proposed Pigeon-inspired Optimization (PIO) iterative process effectively mitigates this issue.

While swarm intelligence algorithms are mentioned briefly, there is insufficient exploration of how they enhance robustness and improve parameter selection within the context of DL modeling.

Although the introduction suggests the framework's potential independence of tree species, there is a lack of discussion on its adaptability to different ecosystems.

5.     Materials (Section 2):

The description of the test site lacks information on its ecological diversity, which could impact the generalizability of findings.

Test Site (Section 2.1):

The emphasis on elevations as natural boundaries is redundant, as it's already established that the study area is demarcated by higher elevations.

While climate information is vital, the extensive details might be more suitable in an environmental science context than a general readership.

Field-Based AGB Calculation (Section 2.2):

The origin of the Forest Resources Planning and Design Survey data isn’t explicitly mentioned, introducing ambiguity regarding its reliability.

Feature Extraction (Section 2.4):

The rationale behind selecting specific vegetation indices and biophysical parameters could be clearer.

Methodology (Section 3):

Construction of Feature Set (Section 3.1):

The rationale behind creating specific feature groups (Groups I to V) is not explicitly stated.

Development of AGB Model (Section 3.2):

The pigeon-inspired optimization (PIO) algorithm is introduced without a clear justification for its choice over other optimization methods.

Comparisons with other optimization algorithms are absent.

AGB Model Validation (Section 3.3):

The BiLSTM model is introduced as a baseline, but the rationale behind choosing it as the baseline model isn't explained.

Additional evaluation metrics, such as precision or recall, could provide a more comprehensive assessment.

References

[1] CNN Based Automated Weed Detection System Using UAV Imagery

[2] Unmanned aerial vehicle and artificial intelligence revolutionizing efficient and precision sustainable forest management

[3] Deep Learning Based Supervised Image Classification Using UAV Images for Forest Areas Classification

Comments on the Quality of English Language

 Extensive editing of English language required.

Round 2

Reviewer 1 Report

Comments and Suggestions for Authors

Great job with the revision. I only a few more concerns:

1. I still feel the paper is too technical, try to proofread again and make sure you describe technical terms clearly

2. Too much results and long tables, consider appending some.

3. Emphasize more in the discussion the uncertainty associated with your "framework" especially in the output AGB map 

Comments on the Quality of English Language

Since many edits since first revision, I would suggest several rounds of proofreading

Reviewer 2 Report

Comments and Suggestions for Authors

Dear Authors

I have observed that all the comments have been addressed satisfactorily.

Author Response

Thank you for your comments.